# Network architecture of transcriptomic stress responses in zebrafish embryos

Kaylee Beine[1], Lauric Feugere[2], Alexander P. Turner[3], Katharina C. Wollenberg Valero[1,4*]

1 School of Biology and Environmental Science, University College Dublin, Dublin, Ireland,
2 Département de Biologie, Chimie et Géographie, University of Quebec at Rimouski, Rimouski, Canada,
3 Department of Computer Science, University of Nottingham, Nottingham, United Kingdom, 4 Conway Institute, University College Dublin, Dublin, Ireland

* katharina.wollenbergvalero@ucd.ie

## Abstract

Protein-protein interaction (PPI) network topology can contribute to explaining fundamental properties of genes, from expression levels to evolutionary constraints. Genes central to a network are more likely to be both conserved and highly expressed, whereas genes that are able to evolve in response to selective pressures but expressed at lower levels are located on the periphery of the network. The stress response is likewise thought to be conserved and its associated genes highly expressed, however, experimental evidence for these patterns is limited. Therefore, we examined here whether the transcriptomic response to two environmental stressors (heat, UV, and their combination) is related to PPI architecture in zebrafish (*Danio rerio)* embryos. We show that stress response genes are situated more centrally in the PPI network. The transcriptomic response to heat was located in both central and peripheral positions, whereas UV response transcripts occupied central to intermediate positions. Network position was further linked to the magnitude of fold changes of genes and number of their associated phenotype GO terms. Across treatments, differentially expressed genes in different parts of the network affected identical phenotypes. Our results indicate that the zebrafish stress response is considered conserved but also have stressor-specific aspects. These properties can aid in better understanding the organismal response to diverse and co-occurring stressors. Given the speed of contemporary changes in aquatic ecosystems, our approach can aid in identifying novel key regulators of the systemic response to specific stressors.

## Author summary

Understanding how genes interact with each other within networks can reveal important insights into cellular activity and evolution. Generally, genes that play a central role in networks tend to be both conserved and highly active, while genes on the outskirts are more adaptable but less active. We investigated how

Data availability statement: Data for this study are provided as supplementary excel spreadsheet. Code is deposited on GitHub: https://github.com/KBeine/Network_architecture_zebrafish_PPI/tree/main.

Funding: This work was funded by the Royal Society (RGS\R2\180033) to KCWV. KCWV and KB acknowledge funding by the European Union (ERC, MolStressH2O, 101044202). Views and opinions expressed are however those of the author(s) only and do not necessarily reflect those of the European Union or the European Research Council Executive Agency. Neither the European Union nor the granting authority can be held responsible for them. The funders had no role in study design, data collection and analysis, decision to publish, or preparation of the manuscript. KB received salary via ERC, MolStressH2O, 101044202.

Competing interests: The authors have declared that no competing interests exist.

zebrafish genes respond to different stressors (heat, UV radiation, and their combination), and how these responses relate to their positions in the network. Our findings show that genes involved in stress responses are central in the network. Heat stress additionally causes activity changes in genes at the edges, while UV stress additionally influences intermediate genes. Despite different stressors affecting genes in various network locations, the impacted genes lead to similar physical traits in the zebrafish. This suggests that while the stress response is largely conserved, there are also unique aspects depending on the specific stressor. Our approach can help identify new key genes that regulate how zebrafish respond to environmental changes, providing valuable insights for understanding how aquatic organisms adapt to rapid environmental shifts.

## Introduction

An organism's capacity to perceive and adjust to environmental challenges plays a crucial role in upholding functional cellular homeostasis [1]. It also may align to the potential of species to respond to climate change by either acclimatizing to stressful conditions or by shifting their geographical distribution to more favorable habitats [2]. In recent years, there has been an increased focus on ectothermic organisms, which are heavily influenced by temperature dynamics [3,4]. In addition to more frequent, abnormally high temperatures during heat waves, organisms must also deal with stress induced by other aspects of the abiotic environment, such as UV radiation (UV) experienced from sunlight [5], which is also expected to become more intense in the near future [6]. However, aquatic organisms may show more complex responses to combined stressors due to interactive effects [7,8]. For instance, UV and heat can act in combination to synergistically or antagonistically impact fish biology [9–12]. UV has been shown to affect biological processes in ectotherms from fish [13], to insects [14], to amphibians [15] including growth, homeostasis, and DNA damage and repair. This has the potential to result in negative consequences for the performance and fitness of the organisms. These molecular responses to stressors have the potential to result in negative consequences for the performance and fitness of the organisms. For example, heat strongly affects the integrity of cellular structures [16], denatures DNA repair proteins [5], and increases ROS production [17] in fish. Similarly, UV can cause damage to proteins, lipids, and DNA in fish [5,10,18]. Fish embryos are more vulnerable to heat and UV than their adult counterparts, as developmental functions are impacted by mutations and heat stress [10,13]. Heat stress combined with UV during early-life development can increase malformations and mortality and so reduce the fitness of fish, which lay eggs in shallow, often warm, UV-exposed waters [5,10,12,19,20]. Both ultraviolet A (UVA, 315–400 nm) and B (UVB, 280–315 nm) cause damage to macromolecules [5,10,21], but UVA (and blue light) can activate repair mechanisms to reverse DNA lesions [21–23]. As such, photorepair during UVA following UVB exposure can partially rescue malformed zebrafish embryos, confirming the photorepair capacity of this species [12,23]. In addition, sublethal heat stress

during development can have a hormetic effect, protecting zebrafish embryos grown under high temperatures against further damage through UV [12]. In summary, fish embryos represent an ideal model to investigate whether naturally co-occurring stressors such as UV and heat stress activate conserved vs. unique components of the genome network.

One central role in cellular acclimation to short- or long-term environmental stressors lies in the modulation of gene expression, where extensive regulation takes place both at the transcriptional and post-transcriptional levels [1]. The control of gene expression in response to a stressor is both tightly regulated and reversible, depending on the type of stress and the organism [24]. For instance, warm temperatures during embryonic development cause later-life effects on the transcriptomic network organization in laboratory lines of multiple fish species, including zebrafish, increasing its network entropy [13,25,26]. The stress response is regarded to be adaptive, and depending on the organism and the type of stress, can either be generically shared by multiple stressors or be specific to a particular stressor [24,27]. Any stressor-specific response may trigger deleterious trade-offs within multiple stressor environments, which is why the general stress response is thought to have an evolutionary conserved component [24,28].

Genetic networks, such as protein-protein interaction (PPI) networks, describe functional interactions of proteins within the cell. Genetic networks are often used in human research [29,30]. Additional studies have focused on network evolution in fungi [31–34] and bacteria [35,36], with very few studies pertaining to vertebrate animals (but see [27,37] for examples of climate adaptation-related networks). However, a few general patterns are emerging: genes towards the center of the network are most conserved, and those positioned intermediately have the highest number of connections with other nodes, whereas both constraint and pleiotropy are lowest at the network periphery (also see [38,39]). Networks were classified based on three characteristics to show and derive meaning from the positioning of proteins within the PPI network: Neighborhood Connectivity (NC, with the highest values characterizing nodes located intermediately in the network and with the highest number of edges to other nodes), Average Shortest Path Length (ASPL, with the highest values characterizing nodes peripherally located in the network and with few connections to other nodes), and Betweenness Centrality (BC, with highest values characterizing nodes central to the network) [34]. While these properties pertain to adaptive processes accumulated over multiple generations, they nonetheless also affect gene expression, as functional connections in a PPI denote real-time interactions between gene products and can thus link evolution to function. Networks can explain such emergent properties which are not apparent when studying genes or pathways individually [40]. Highly-expressed proteins are under strong natural selection for correct protein folding and function, and thus evolve slowly, a phenomenon known as the "E-R anti-correlation" [31,34]. These genes, located centrally in the interactome, are functionally constrained as they are essential for cell survival [34,41,42], and show a lower evolutionary rate [43]. Gene expression is lower in highly pleiotropic genes, a phenomenon dubbed the "cost of complexity" [44,45]. The number of PPI is lower in proteins with higher evolutionary rates [46,47]. In vertebrates, a set of ~1000 genes identified to adapt to climatic gradients is to a large part differentially up- or downregulated (38%, and 22.5%, respectively) in response to short-term environmental stressors such as heat or cold stress [27]. Of these genes, 25% are also "wider definition" housekeeping genes with intermediate levels of expression [27,48]. It is therefore of interest to study the location of genes involved in the stress response within PPI networks, particularly with respect to combined stressors, in order to better understand the interplay between expression levels, stress response, and network position.

In this paper, we analyze a transcriptomic dataset previously generated from developing zebrafish embryos [12] exposed to heat stress (TS) and UV radiation (UV). Our analysis includes three treatment comparisons: one for each stressor against a control, and one comparing UV following TS to UV exposure alone. Four specific hypotheses are tested. Hypothesis (1) is that genes linked to the stress response, in terms of their Gene Ontology (GO), cluster centrally within the zebrafish PPI network, reflecting both a high degree of evolutionary constraint and the potential for high gene expression expected under such a central, conserved stress response. Hypothesis (2) is that this pattern also applies to genes differentially expressed in response to two distinct stressors (TS and UV). Under hypothesis (3), we expect that a subset of genes are expressed only when UV follows TS (synergistically), and others are only expressed under each

single stress condition (antagonistically), expected to be located in peripheral regions in the network. Hypothesis (4) anticipates that phenotypic outcomes (such as gene ontologies, and genes' links to phenotypes and diseases), along with the relative changes in gene expression, are influenced by their positions within PPI network. Genes located in hub or intermediate positions of the PPI are expected to show both higher expression levels and evolutionary constraint, and due to pleiotropy may have significant phenotypic effects. In contrast, peripheral genes, which generally have lower levels of expression and constraint, may exhibit more variable phenotypic effects and larger fluctuations in expression changes.

## Results

### Stress response GO term genes are more central to the PPI network

Based on classification using BC, NC, and ASPL, the zebrafish interactome (Fig 1A) are composed in majority of peripheral (P) nodes (62.1%), followed by intermediate (I) (35.7%) and hub (H) (2.13%) nodes. The stress GO term genes had similar overall proportions, but were located with less peripheral (58.9%), but more intermediate (37%) and central (4.19%) node positions within the interactome (pie chart percentages found in *Table K in* S1 File). Fig 1B shows the pairwise comparison between network statistics of the interactome compared to the placement of the stress GO term genes within it. The stress GO term genes occupied a significantly lower ASPL (Kruskal-Wallis test H-statistic, KW-H: 122.454, $p<0.001$, with a small effect size), lower but not significantly different NC, and significantly higher BC than the interactome (KW-H: 127.626, $p<0.001$, with moderate effect size) (*Table A in* S1 File).

### Experimental stress response DEGs are more central to the interactome

Differentially expressed genes (DEGs) from the three different treatment comparisons TS, UV and TS->UV were visualized and further subsetted based on a Venn diagram (Fig 2). Overall, 71 genes were shared between all three treatment comparisons (TS, UV, TS->UV). Another 187 genes were shared between UV and TS->UV, 121 genes were shared between TS and TS->UV, and 135 genes were shared antagonistically in both TS and UV but not TS->UV. In addition, 134 genes were only differentially expressed in TS but in no other combination, 1720 genes were only differentially

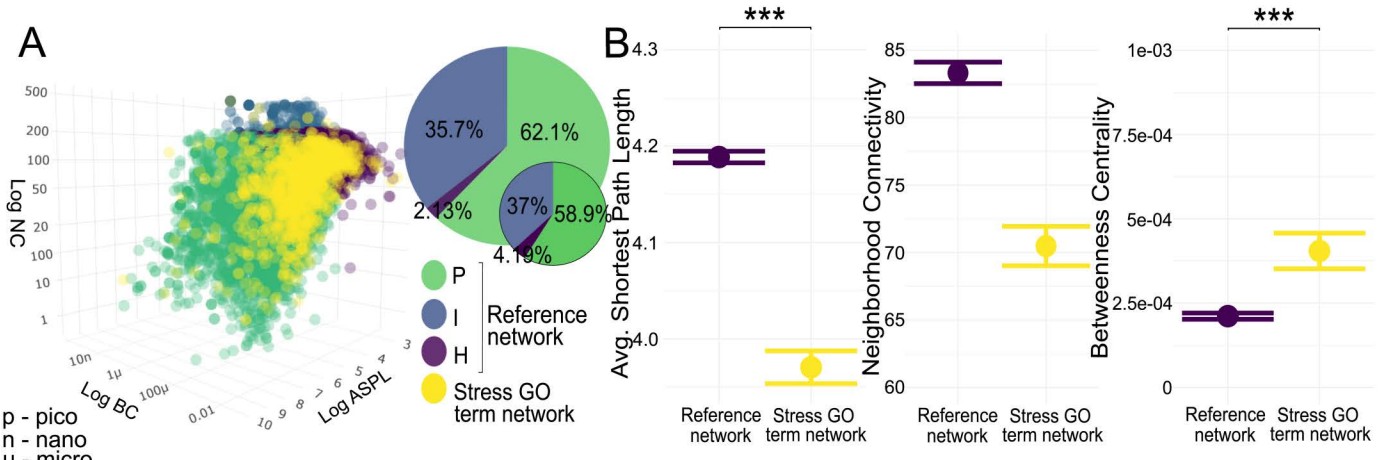

**Fig 1. The position of stress GO term genes within the zebrafish interactome.** (A) Distribution of interactome nodes within network parameter space represented by Average Shortest Path Length (ASPL), Neighborhood Connectivity (NC), and Betweenness Centrality (BC), with node categories in green (Peripheral nodes, P), blue (Intermediate nodes, I), and purple (Hub nodes, H). Stress GO term genes are shown in yellow. Inset pie charts represent node category distribution (large: interactome; small: stress GO term genes). (B) Means with standard error plots of ASPL, BC and NC for the two gene sets. Significance levels of pairwise Mann-Whitney U tests are indicated with asterisks as $p<0.0001$: ***; $p<0.001$: **; $p<0.05$: *. p - pico, u - micro, n - nano.

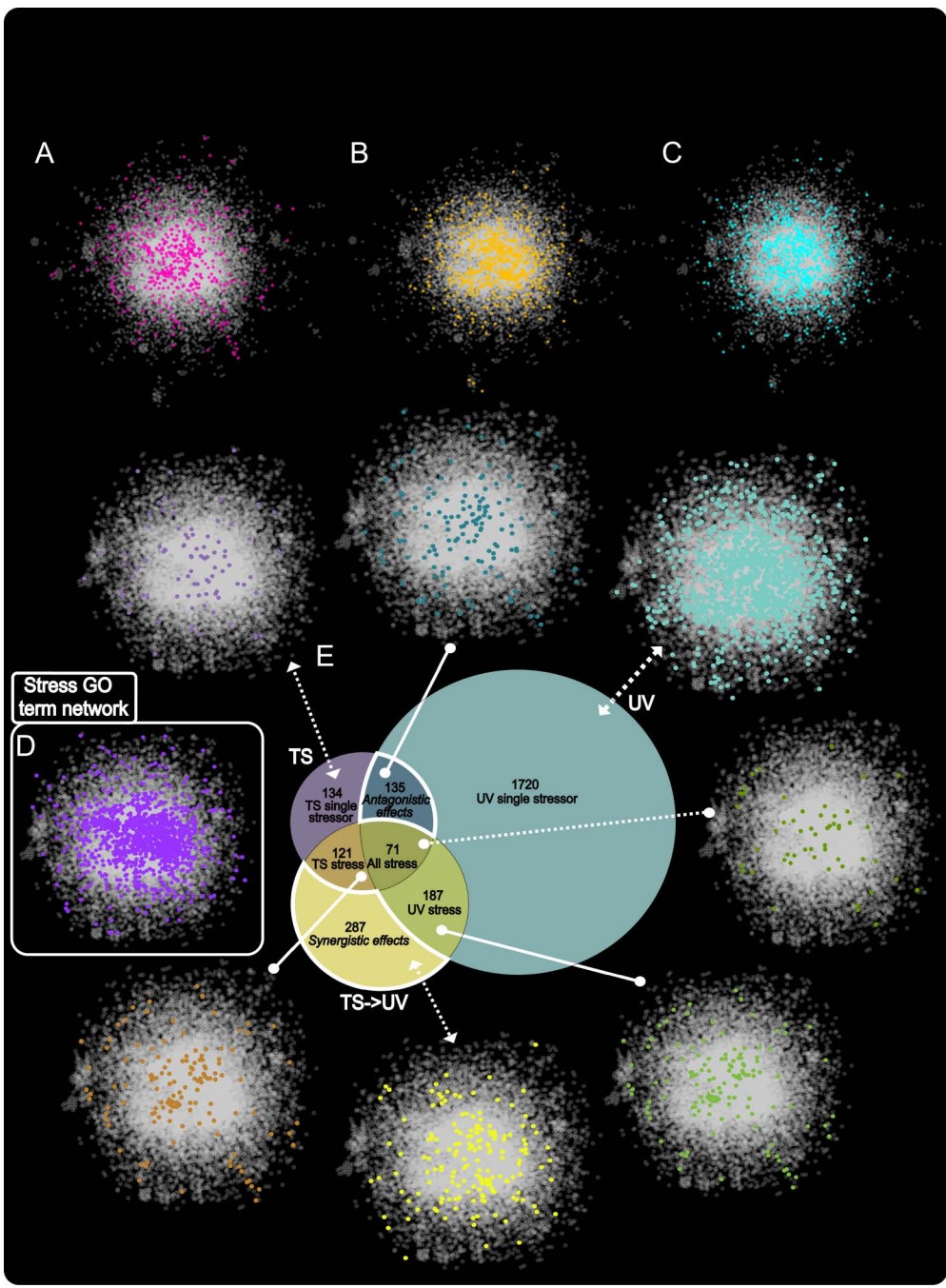

**Fig 2. Visualization of significant DEGs in response to thermal stress, UV exposure, and UV exposure following thermal stress, highlighting differences (white) and commonalities (gray).** The networks displayed in reference to the Venn diagram show the positioning of the genes relative to the zebrafish interactome and the stress GO term network. A - thermal stress DEGs, B - UV stress DEGs, C - Thermal stress followed by UV stress DEGs, D - Stress GO term genes, E - Venn diagram with corresponding single stressors/combination stressors as isolated by overlapping genes. All stressors (dark green), TS-responsive DEGs (brown), antagonistic DEGs (dark blue), TS-only DEGs (lilac), UV-only DEGs (light blue), and synergistic DEGs (yellow).

expressed in UV but no other combination, and 287 were only differentially expressed in TS->UV synergistically, but not in the individual TS and UV treatment comparisons.

Overall, the relative position of all experimental DEGs across treatment comparisons TS, UV and TS->UV significantly differed from that of the zebrafish interactome (ASPL KW-H: 20.778, $p < 0.001$; NC KW-H = 19.837, $p < 0.001$; BC KW-H: 19.795, $p < 0.001$, with small effect size for all predictors, Fig 3 and Table B in S1 File). Pairwise *post hoc* tests (*Tables C-E in S1 File*) revealed that TS did not have a significantly different ASPL compared to the interactome but had both a lower

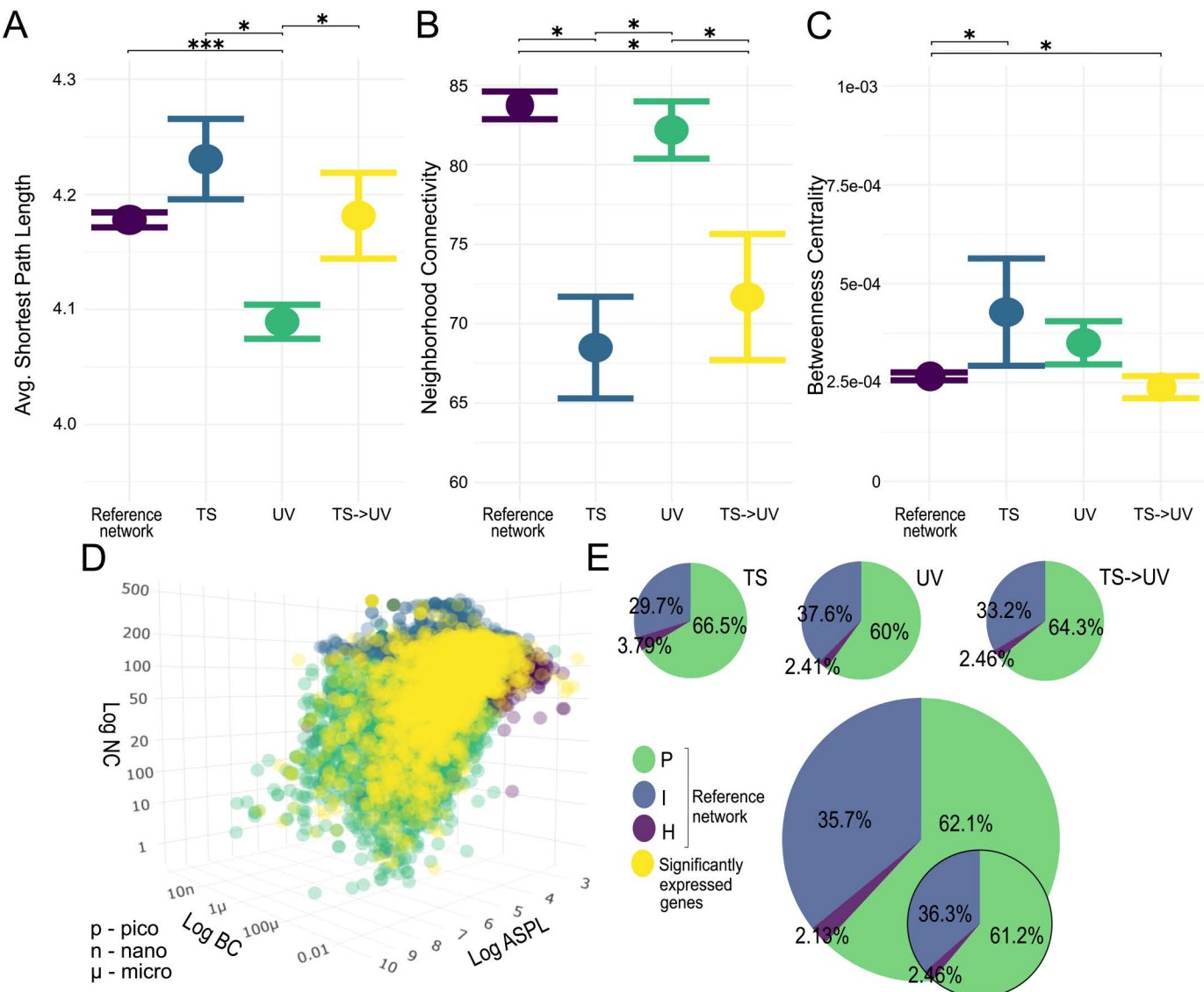

**Fig 3. The position of experimental DEGs within the zebrafish interactome.** (A-C) Means with standard error plots of ASPL, BC and NC for each gene set's network dimensions with significance levels of pairwise Mann-Whitney U tests indicated with asterisks as $p < 0.0001$: ***; $p < 0.001$: **; $p < 0.05$: *. The reference network is the interactome and TS, UV and TS->UV represent treatment comparison DEGs. (D) Distribution of interactome nodes within network parameter space represented by NC, ASPL, BC, with node categories shown in green (P nodes), blue (I nodes), and purple (H nodes). Experimental DEGs are shown in yellow. (E) Inset pie charts represent node category distribution (large: interactome; smaller: experimental DEGs, smallest: experimental DEGs per treatment comparison).

NC (W = 630.619, *p*-adj. = 0.01) and higher BC (W = -525.45, *p*-adj. = 0.02). Correspondingly, it had 4.4%, (n = 20 genes) more P nodes, 6%, (n = 27) fewer I nodes, and 1.33%, (n = 6) more H nodes compared to the interactome (Tables C-E in S1 File). The network of DEGs from the UV treatment comparison had a significantly lower ASPL (W = 420.675, *p*-adj. < 0.001), and no significant difference in NC and BC. As expected, it had 2.1% (n = 44) fewer P nodes, but also 1.9% (n = 40) more I and 0.33% (n = 6) more H nodes (pie chart percentages found in Table K in S1 File). The combined TS->UV treatment comparison did not have a significantly different ASPL compared to the interactome, but a lower NC (W = 453.921, *p*-adj. = 0.04), and higher BC (W = -481.9, *p*-adj. = 0.008). In accordance with this, the number of P nodes was 2.2% (n = 15) higher, the number of I nodes 2.5% (n = 17) lower and there were 0.33% (n = 2) more H nodes (Tables C-E in S1 File).

### The topology of DEGs shows the same pattern relative to the expressed portion of the network

The genes expressed during the experiment differed in their network topology from the interactome. The number of H nodes expressed in this part of the interactome was 1.88% (n = 205 genes) lower, and most importantly, the number of P nodes was 36.4% (n = 3987) lower in favor of 38.3% (n = 4194) more I nodes than the zebrafish interactome. Network statistical parameters of experimental DEGs within this expressed network significantly differed between treatment comparisons in ASPL (KW-H: 13.226, *p*-value: 0.004) and NC (KW-H: 20.253, *p*-value: <0.001), but not BC (KW-H: 6.297, *p*-value: 0.098, with small effect size for all predictors, *Fig 4* and *Table D in* S1 File). Consequently, no pairwise *post hoc* tests for BC were performed. In addition, while ASPL was a significant predictor in the overall analysis, treatment comparisons did not significantly differ in ASPL in *post hoc* tests (*Table G in* S1 File). However, DEGs from the TS treatment comparison had significantly lower NC (W = 504.88, *p*-adj. = 0.009) than the expressed network (*Table G in* S1 File). Compared to the expressed reference network, only one more H node was activated (0.2%), but I nodes decreased by 7.2% (n = 33) in favor of 7.07% (n = 33) more P nodes (*Table K in* S1 File). DEGs from the UV treatment comparison did not differ in ASPL, NC or BC compared to the expressed network. However, one more H node was activated (0.059%, n = 1), and 1% (n = 23) fewer P nodes were activated in favor of 1% (n = 21) more I nodes (*Table K in* S1 File). DEGs from the TS->UV treatment comparison had significantly lower NC than the expressed network (W = 362.267, *p*-adj. = 0.039; *Fig 4B* and *Table H in* S1 File). There were only 2 fewer H nodes (-0.25%), but a decrease in I nodes of 3.3% (n = 22) and a concomitant increase in P nodes of 3.6% (n = 24, *Table K in* S1 File).

### Combined stressors show unstructured positioning while stressor-unique DEGs show centralization within the network

Neither synergistic (only present in TS->UV) or antagonistic DEGs (present in both TS or UV comparisons but not in TS->UV) differed by node statistics (S1 Fig, Table I *in* S1 File). Nonetheless, the synergistic gene set had 3.7% (n = 11) more P genes and 3.5% (n = 10) fewer I genes compared to the interactome (Table K *in* S1 File). The antagonistic gene set had 7.9% (n = 11) fewer I nodes and 6% (n = 8) more P nodes relative to the interactome (Table K *in* S1 File). In addition, network statistical parameters did not differ significantly between DEGs in the TS treatment comparison and the interactome (Fig 5 and Table J *in* S1 File), but TS had +0.92% (n = 8) more P nodes and 6.7% (n = 9) fewer I nodes (Table K *in* S1 File). However, UV as a single stressor activated significantly more 2.7%, (n = 8) I nodes and significantly fewer P nodes 2.7%, (n = 5) than the interactome, supported by both lower ASPL (H = 12.416, p < 0.001, with a small effect size, Fig 5 and Table J *in* S1 File) and higher BC (H = 16.584, p < 0.001, with a small effect size, Fig 5 and Table J *in* S1 File).

### Differences in DEG positions relate to changes in phenotype between treatment comparisons, and are reflected in fold change

Phenotypic outcomes varied across the three treatment comparisons. In the UV treatment comparison, embryos were generally shorter and had more defects. TS and TS->UV treatment comparisons were similar to one another in terms

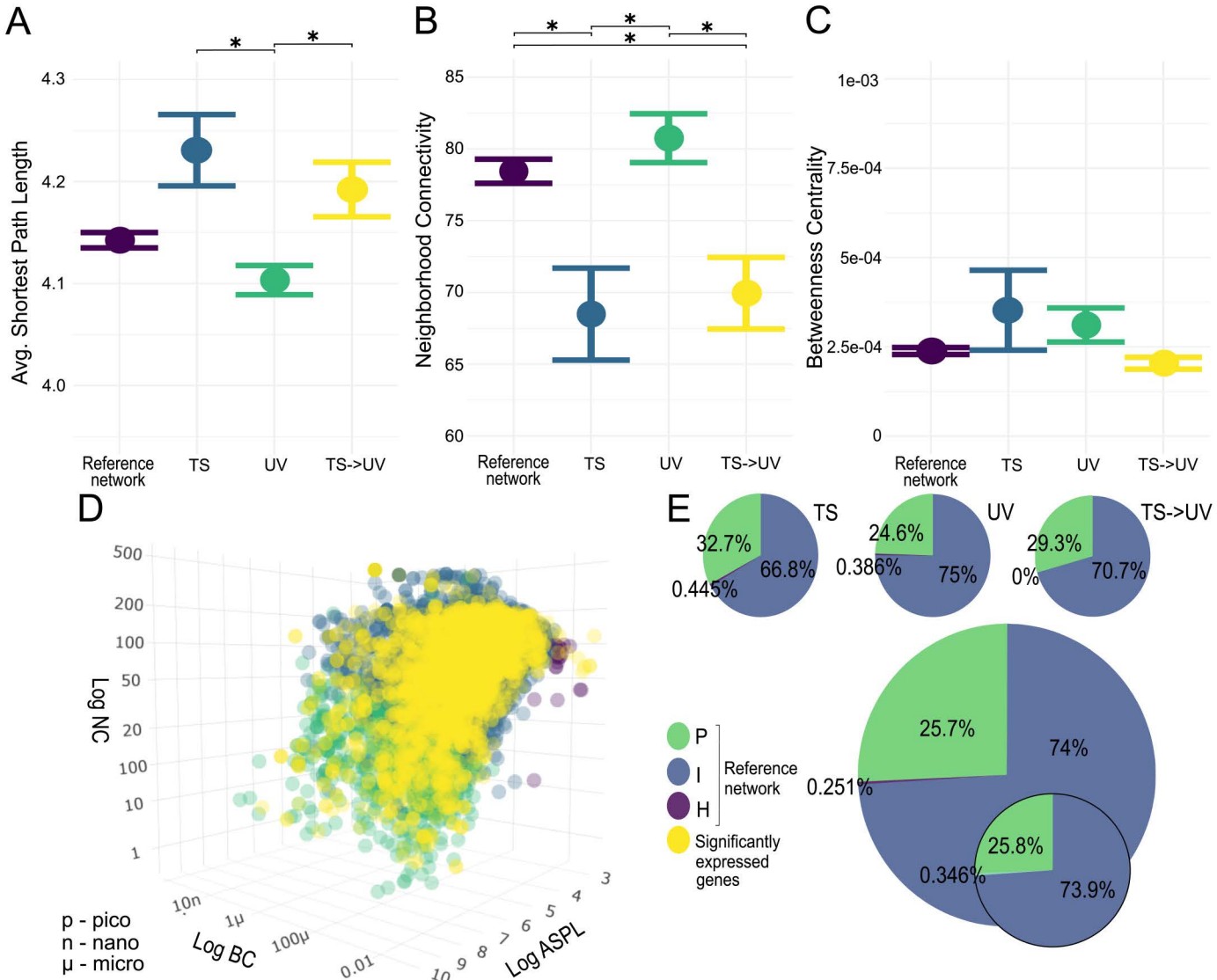

**Fig 4. The position of experimental DEGs within the expressed network.** (A-C) Means with standard error plots of ASPL, BC and NC for the compared gene sets' network dimensions with significance levels of pairwise Mann-Whitney U tests indicated with asterisks as $p < 0.0001$: ***; $p < 0.001$: **; $p < 0.05$: *. The reference network is constructed from genes expressed during the experiments and TS, UV and TS->UV represent treatment comparison DEGs. (D) Distribution of expressed network nodes within network parameter space represented by NC, ASPL, and BC, with node categories shown in green (P nodes), blue (I nodes), and purple (H nodes). Experimental DEGs are shown in yellow. (E) Inset pie charts represent node category distribution (large: expressed genes; smaller: experimental DEGs, smallest: experimental DEGs per treatment comparison).

of (compared to UV) longer embryos with fewer defects (Fig 6, S1 Dataset, *Co-variation of DEG network*). The sum of defects and shortest embryo length (SEL) covaried with ASPL and NC, while BC covaried with the sum of defects and SEL for UV and TS->UV, but not in TS. In TS and TS->UV, defects were low, SEL was high, with high NC and low ASPL. In UV, defects were high, SEL was low, NC high and ASPL low. BC was high in TS, lower in UV, and lowest in TS->UV.

We explored significant biological processes, phenotypes, and diseases associated with DEGs' network positions across treatments (Fig 7, for complete results see S1 Dataset, *4. Gene nodes*). Six H-node biological processes, including carboxylic acid metabolism and response to oxidative stress, were common across all treatments. H nodes of TS and UV

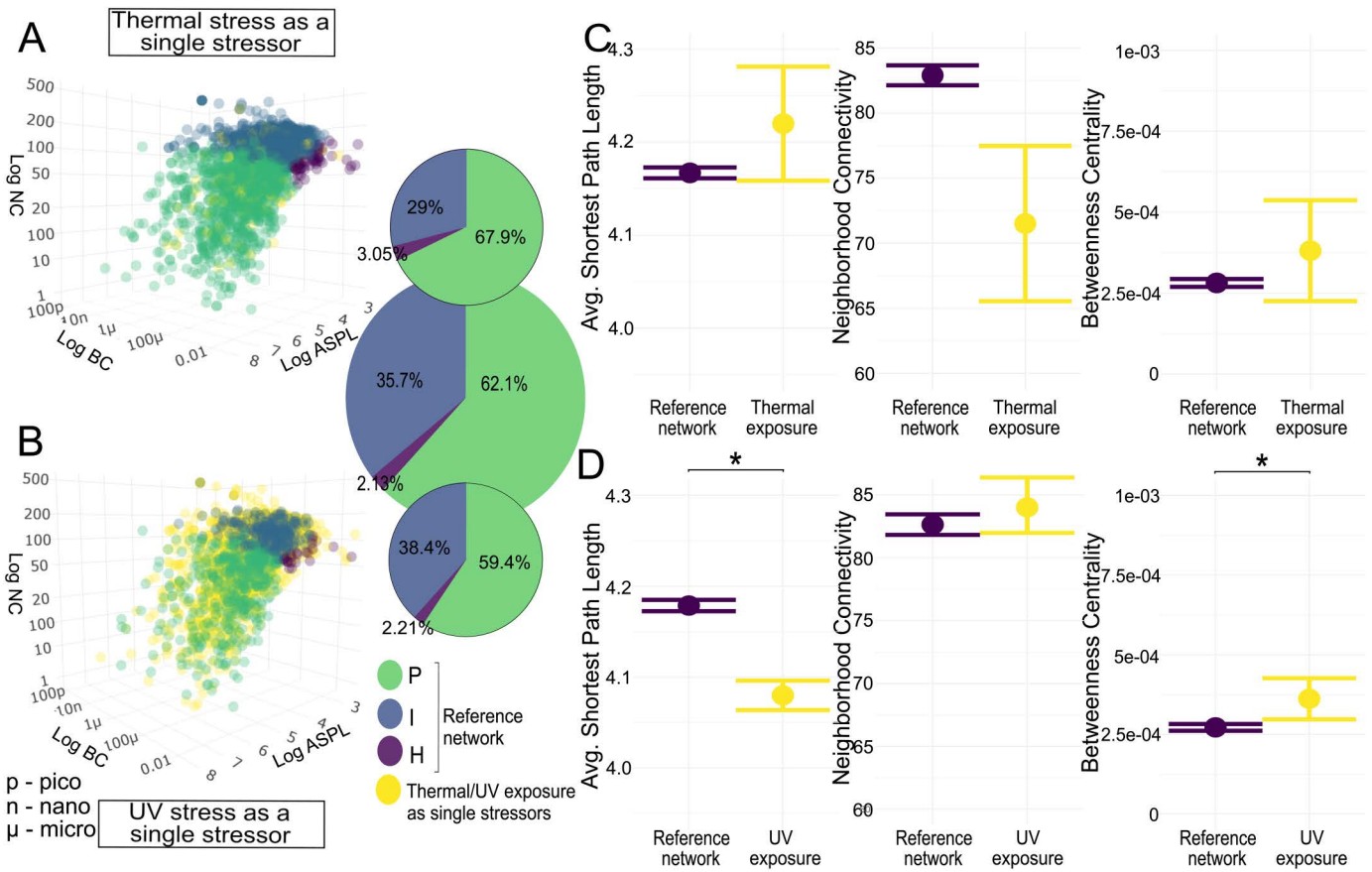

**Fig 5. The position of the unique DEG responses to heat and UV treatment comparisons within the zebrafish interactome.** (A) Distribution of interactome nodes within network parameter space represented by NC, ASPL, and BC, with node categories shown in green (P nodes), blue (I nodes), and purple (H nodes). (A) TS-unique DEGs and (B) UV-unique DEGs are shown in yellow, respectively. Inset pie charts represent node category distribution (large - interactome; small/top - TS, small/bottom - UV -unique genes). (C, D) Means with standard error plots of ASPL, BC and NC for the interactome and TS and UV - unique gene sets' network dimensions. Significance levels of pairwise Mann-Whitney U tests are indicated with asterisks $p < 0.0001$ ***; $p < 0.001$ **; $p < 0.05$ *.

treatments shared amino acid and small molecule metabolic processes. Fourteen I-node processes, like ADP hydrolysis and glycolysis, were shared between TS and TS->UV, with further unique processes specific to each treatment. Twelve biological processes in the P node category across all treatments included cell development and differentiation, muscle development between TS and TS->UV, and eye development and morphogenesis shared between UV and TS->UV. Tissue, organ development, and morphogenesis were processes shared between H and P nodes across UV and TS->UV. Processes shared between H and I nodes, including carbohydrate catabolism, were associated with UV (Fig 7, S1 Dataset, *Biological processes*).

No phenotype GO term was exclusive to H nodes, but I nodes across UV and TS->UV involved abnormalities in the central nervous system and yolk appearance. P node-specific GO term phenotypes included disrupted thigmotaxis and abnormalities in brain volume, spinal structure, and osteoblast accumulation in TS and TS->UV treatments. Decreased eye and head size, and edematous pericardium were common GO terms across treatments but varied in node category, such as H nodes in TS and I and P nodes in UV and TS->UV (Fig 7, S1 Dataset, *Phenotype*).

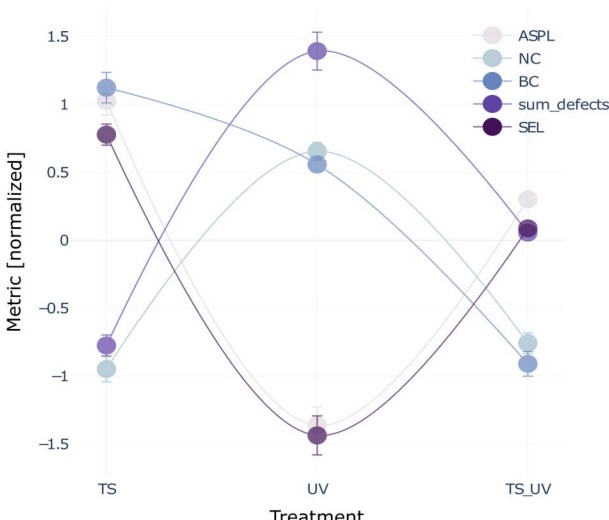

**Fig 6. Co-variation of DEG network parameters and phenotypic outcomes.** Normalized average values per treatment comparison (with error bars) for interactome-derived network parameters ASPL, NC and BC of DEGs per treatment comparison and phenotypic parameters embryo length and defects. ASPL, NC, sum of defects and SEL co-vary among treatment comparisons while BC shows a different pattern of being highest in TS, lower in UV, and lowest in TS->UV DEGs.

H-node-linked diseases included diabetic retinopathy with TS and UV, while TS and TS->UV DEGs in H nodes included diseases like melanoma and pancreatic cancer. I node-specific diseases common to TS and UV included sciatic neuropathy, with additional diseases like neuromuscular disease and glycogen storage disease linked to TS and TS->UV. (Fig 7). Diseases associated only with the P node category in TS and TS->UV treatments included heart disease, and osteogenesis imperfecta. Spanning multiple node categories, Type 2 diabetes mellitus was linked to all treatments: P nodes in TS and UV, and H nodes in TS->UV. Diseases common to at least two treatments included cardiomyopathy, cataract, and Alzheimer's disease. UV-specific diseases across multiple node categories included breast cancer, hypertension, and prostate cancer (Fig 7, S1 Dataset, *Diseases*)

No common biological processes or phenotypes were observed between the antagonistic and synergistic effects of TS and UV. Diseases linked to antagonistic DEGs included diabetic retinopathy (P and H nodes) and type 2 diabetes mellitus (P and I nodes). Breast carcinoma was the only disease associated with both effects, with relevant genes found in the H node category.

Scatterplots of squared log fold change (lfc) of DEGs per treatment comparison reveal two distinct categories: one with low squared log fold change and involvement in a high number of phenotypic effects, and one with a high squared log fold change and involvement in a lower number of phenotype GO terms (cumulative sum of gene ontologies/biological processes, ZFIN phenotypes or ZFIN diseases per gene). The majority of I and H nodes are in the first group where the latter group are composed of the majority of P nodes, with the exception of phenotype components and phenotype log fold change, which however covers the smallest range of log fold change values among all comparisons. A few H and I genes comprise notable outliers to this pattern, with higher fold changes combined with lower phenotypic involvement; these are *slc25a4* ($p$-adj. of TS < 0.001, lfc of TS = 2.55, H-node) and *hsc70* ($p$-adj. of TS = 0.052, lfc of TS = 2.28, I-node) in TS, *hsp70.2* ($p$-adj. of UV < 0.001, lfc of UV = 6.47, I-node) in UV, and *slc25a4* ($p$-adj. of TS->UV < 0.001, lfc = 1.46, I-node) and *grin1b* ($p$-adj. of TS->UV < 0.001, lfc = 1.62, I-node) in TS->UV (Fig 8).

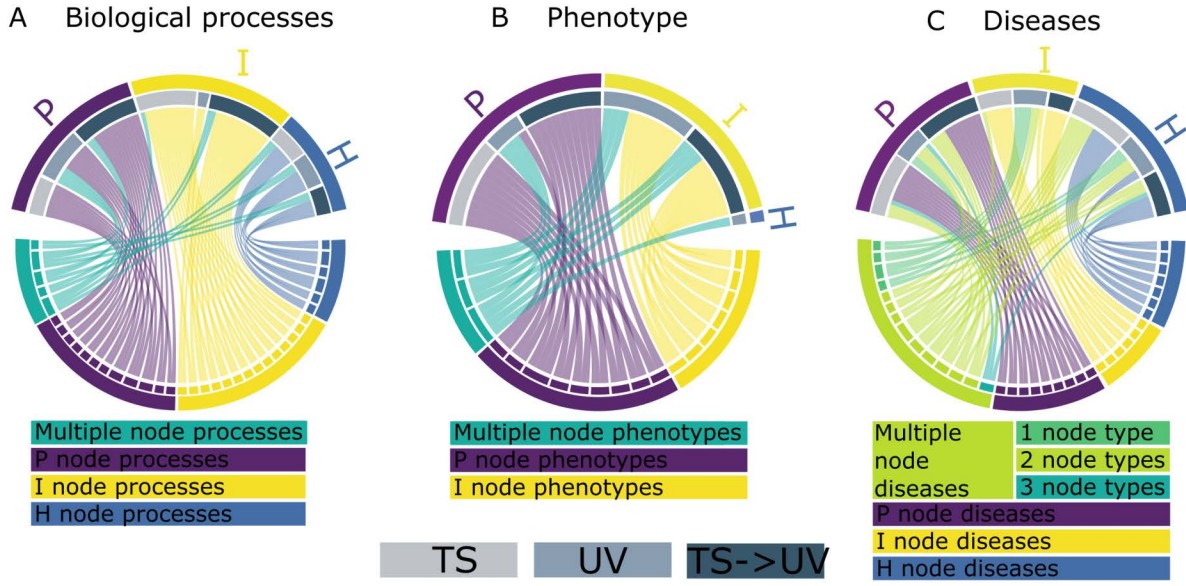

**Fig 7. Shared phenotypic effects of DEGs by treatment comparison and network node category.** (A-C) Chord diagrams showing shared phenotypic properties of DEGs within the zebrafish interactome (bottom half circle) depending on interactome node location per treatment comparison (top half circle: TS - light gray; UV - medium gray - TS>UV - dark gray). Where (A) shows shared Gene Ontologies (only Biological Process), (B) phenotype and (C) disease implications in zebrafish. Blue ribbons indicate phenotypic properties exclusively associated with H nodes, yellow - I nodes, purple - P nodes. In contrast, green shades indicate phenotypic properties common across multiple node categories with turquoise representing all three treatment comparisons; light green representing only two treatment comparison types, and dark green indicating only one treatment comparison type across different node types. Here, for a complete list of significantly enriched terms and genes contributing to them see S1 Datasets, Biological processes; Phenotype; Diseases).

## Discussion

### Stress response genes are concentrated towards the center, but heat and UV also activate different parts of the zebrafish interactome

This study analyzed the network-related aspects of the transcriptomic response to two naturally co-occurring environmental stressors, heat and UV radiation, in zebrafish embryos. Genes with stress-related ontologies were significantly more prevalent in intermediate and central interactome nodes, indicating high evolutionary constraint and robust stress-induced expression [49,50]. In our experiment, 71 genes were consistently expressed across all treatments, indicating a core set of stress-responsive genes mediating a generalized response via multiple pathways [24,51]. A conserved stress response limits the risk of deleterious effects in mismatched environments [24,28].

However, while the general stress response is conserved, specific contexts may lead to variable activation and outcomes depending on the stressor [49]. Experimentally determined DEGs were not consistently more central to the zebrafish interactome; while all treatments activated more hub nodes, many UV-responsive DEGs were intermediate while TS, and UV following heat, increased DEGs in hub and peripheral nodes. Early development begins with gene regulatory networks characterized by a few transcription factors activating many target genes, until autoregulation begins [52] which is then characterized by further expression differences across timepoints and cell types [53]. The genes expressed by zebrafish embryos in our experiment were not representative of the overall zebrafish interactome, but instead expressed more intermediate nodes. Despite this different node distribution, analysis using the expressed portion of the network as reference did not differ from that using the interactome as a reference. The presence of both central and other network components activated by stress treatments reflects a trade-off between evolutionary robustness and stressor-specific

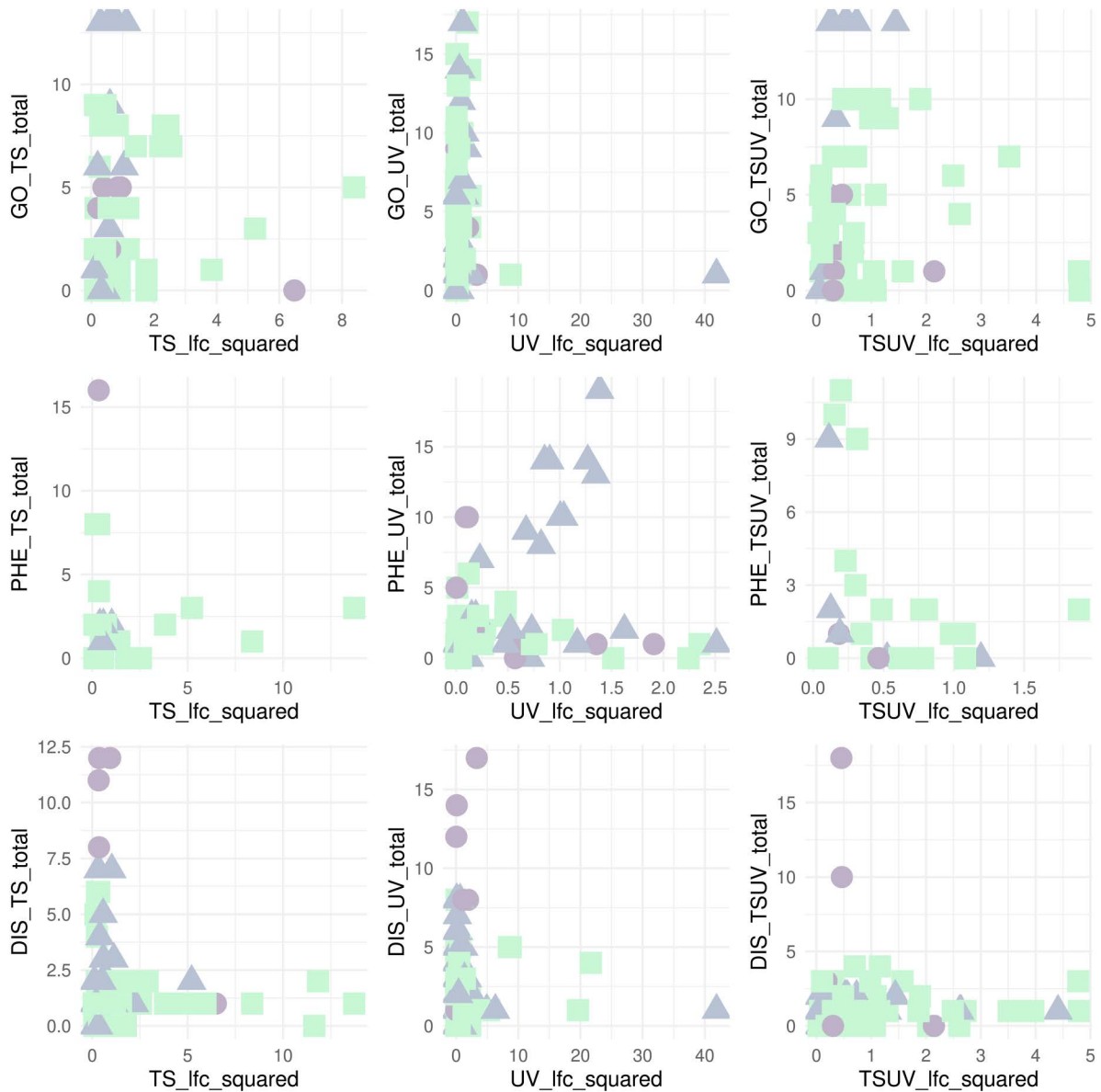

**Fig 8. Phenotypic effects of DEGs by treatment comparison and log fold change depending on network node category.** Scatter plots of squared log fold change in significant DEGs per treatment comparison with node categories represented with different colors and shapes: I-nodes (blue triangles), H-nodes (purple dots) and P-nodes (green squares). GO - Number of Biological Process Gene Ontology terms of genes. DIS - number of disease associations of genes in ZFIN diseases, PHE - number of phenotype associations of genes in ZFIN phenotype. Note that the x and y axes extents differ per each comparison.

gene expression, a requirement for evolvable systems [54], and previously observed in simulations of artificial gene regulatory networks [55,56]. The intermediate position of nodes responding to UV, indicates pleiotropic effects and importance for survival [42,57], and may reflect that UV activates a large number of stress signaling pathways, as well as cell cycle control and apoptosis processes [23,58]. Our results indicate that different environmental stressors engage with network architecture through distinct topological strategies. Thermal stress (TS) and the combined treatment (TS->UV) activated genes with high betweenness centrality (BC) but lower connectivity, suggesting a bridging role across central

Biology

modules. This implies a more dispersed response, where select genes coordinate communication between functionally distinct regions of the transcriptome. In contrast, UV-responsive genes exhibited significantly lower Average Shortest Path Lengths (ASPL) but no significant differences in Neighborhood Connectivity (NC) or BC. This pattern is indicative of both global and modular, internally cohesive activation, which could potentially reflect targeted engagement of tightly connected pathways related to DNA repair and apoptosis. These contrasting strategies - cross-module integration versus modular activation - highlight how transcriptomic stress responses are not only shaped by the nature of the stressor, but also by the underlying network topology through which they are enacted. Further exploration is needed to determine if this response to the UVB/UVA photorepair challenge indicates repair or damage processes, or both [12,23,59,60]. Thermal stress followed by UV exposure (TS->UV compared to UV only) resulted in the second highest number of differentially expressed genes after the UV treatment, suggesting an additive or synergistic effect of sequential stress exposures to both stressors that is not caused purely by the presence of UV. However, these genes synergistically activated in TS->UV did not occupy particular positions in the network, not supporting the hypothesis that genes responding to combined stressor events would be more central. There was also no influence of network topography to genes expressed antagonistically in both TS and UV, refuting the idea that these stressor-unique responses would be more peripheral, instead stressor-unique DEGs from the UV exposure showed centralization within the network. This could perhaps be that UV is considered a mutagen and therefore affects genes that are essential to life and therefore an exposure of UV could target genes that are central to the network.

**Treatments share phenotypic effects unique to node categories**

Within node categories, different stressors further modified the same phenotypic components. Low defect scores and greater embryo length in TS and TS->UV corresponded with low ASPL and high NC, while UV-treated embryos showed lower BC and increased defects. This suggests that both modular cohesion (via NC, in intermediate network positions) and cross-module integration (via BC, in central positions) contribute to mitigating stress effects. These trends align with our fold change analysis, where genes with high pleiotropic impact - mostly I or H nodes - exhibited lower expression shifts but were more phenotypically influential. Together, these results highlight the role of network topology in modulating phenotypic resilience during development. In all treatments, hub nodes were linked to oxidative stress response and amino acid biosynthesis, involving genes associated with phenotypic abnormalities such as cancers, which are critical for managing cellular stress when looking at the biological processes associated with DEGs with shared phenotypic effects [61]. Oxidative stress induced by heat and UV is known to lower protein synthesis via the eIF2α and ATF4 pathways [62]. This underscores the systemic relevance of nodes more central in the interactome, which also have a higher disease relevance [63]. In intermediate network positions, all treatments activated nodes related to energy management and the synthesis of DNA/RNA and proteins under stress, with related diseases affecting muscles and nerves. This aligns with the energetic costs of the stress response due to required changes in transcription and translation [64]. In peripheral network positions, all treatments were linked to genes affecting structural development, or influencing responses to touch and proper spine formation, indicating their role in stressor-specific phenotypic outcomes. A bent spine is a common developmental malformation in zebrafish [65] and has, along with an altered behavioral response to stimuli, previously been observed in response to heat and UV treatment in both zebrafish embryos and larvae [12,39,66].

Pre-exposure to heat before UV caused a shift of DEGs to peripheral nodes which was also observed in the heat treatment. This indicates that beyond the central component of the stress response, stressor-specific adaptability could be built-in towards the network periphery [49]. Heat and UV have previously been shown to activate many of the same pathways as when exposed to either separate treatment in plants [67,68]. These peripheral genes were related to various organ systems, indicating systemic impacts on structural and developmental pathways. Thermal stress activates multiple systems, including the innate immune response [69], oxidative stress regulation [17,70], and a generally faster developmental speed [12]. It also induces regulatory processes like nucleotide metabolism, which may explain the hormetic effect observed, where thermal pre-treatment

either repairs or reduces defects from subsequent UV exposure [12,71–73]. Evidence for a hormetic effect is reflected in the phenotypic outcomes, specifically the lowered occurrence of defects in the thermal stress followed by UV treatment comparison compared to UV alone. The timing of stressors plays an important role in the outcome, for example, the combined impact of two or more stressors can be greater or less than the sum of their individual effects [74]. This shows the importance of simultaneous compared to sequential stressor stimuli. Whereas sequential stressors can cause a tolerance or adaptive response, the hormetic effect, that could mitigate the impact of a subsequent stressor, depending on the order of exposure [75].

Genes related to morphological structure formation in UV treatments were located in both peripheral and hub nodes, which may again represent specific developmental pathways that are regulated centrally, but with specific phenotypic effects through activation of peripheral nodes [76]. UV-associated diseases involve cell growth and the cardiovascular system, highlighting adverse consequences of nucleic acid damage and cell cycle alteration by UV treatments [77]. Both UV and TS->UV treatments showed DEGs with effects on early embryonic development as associated with intermediate nodes. Across all node categories, DEGs in these treatments further shared decreased head size and pericardial edema phenotypes, which have been shown to be typical of UV damage in zebrafish [10]. In summary, through comparing gene functions within network categories, we could here link specific DEGs to specific phenotypic outcomes, which had not previously been revealed in an analysis of all DEGs together [12].

### Network topology-linked expression fold changes differently affect phenotypes

DEGs had either low fold changes with many phenotypic effects or high fold changes with fewer effects. Intermediate and hub nodes were associated with multiple phenotypes due to pleiotropy, supporting the "cost of complexity" and "E-R anticorrelation" concepts, where highly constrained genes are highly expressed [31,34,44,78–80]. Fold changes in such highly expressed genes may have effects on more essential components of the phenotype, even if they are lower in magnitude. Such expression constraint imposed by pleiotropy has previously been found in fish adapted to different stream temperatures, with high numbers of protein-protein interactions lowering protein expression [81].

Nonetheless, we found a few intermediate and hub genes to constitute outliers to this pattern, showing higher expression changes combined with lower phenotypic involvement. These were the two heat shock protein genes *hsc70* (synonymous with *hspa8*; heat shock cognate 70, in TS) and *hsp70.2* in UV. *hsc70* is a constitutive molecular chaperone maintaining protein homeostasis, active in development [82,83], but is known to be upregulated in response to heavy metal exposure in developing sea urchin embryos [84], and in heat stress-induced transposable element activation in the germline [85]. *Hsp70.2* is orthologous to human heat shock protein 70, with molecular chaperone function and is inducible by heat [83,86]. In treatments with heat stress (TS and TS->UV), *slc25a4* (Solute Carrier Family 25 Member 4) constituted another such outlier. This antiporter translocates ADP into the mitochondrial matrix and ATP back into the cytoplasm [83,87], and so could play a major role in energy availability for the stress response [64]. Lastly *grin1b* in TS->UV (Glutamate Ionotropic Receptor NMDA Type Subunit 1) codes for a glutamate receptor which is involved in synapse plasticity, but also MAPK signaling [83] and is known to affect behavior in developing zebrafish [88]. Overall, the roles of *slc25a4* and *grin1b* in the stress response are little explored, compared to the two heat shock proteins. Comparing fold changes to network position may therefore aid in uncovering novel key regulators of the cellular and physiological systemic response to various stressors.

In summary our results support the idea of the response to different environmental stressors being evolutionarily conserved and central to the zebrafish interactome, but also highlights exceptions to this pattern with distinct roles of hub, intermediate, and peripheral nodes in the stress response associated with magnitude of gene expression and developmental outcomes. Integrating network position and fold changes in response to treatments can help identify novel regulators of stress responses. Since the embryonic environment affects later-life phenotypic plasticity, impacting the individual's ability to acclimate to environmental stressors [25,89,90], research on stress responses in aquatic developing organisms should receive increased attention in the context of climate change affecting aquatic environments.

## Methods

### Ethics statement

All experiments were approved by the Ethics committee of the University of Hull (FEC_2019_194 Amendment 1).

All transcriptomic and phenotypic data used in this study were obtained from [12,66], but reanalyzed to make novel comparisons. A more thorough description of methods used such as animal husbandry and setup of the UV treatment comparisons can be found therein. Briefly, zebrafish embryos (2 to 3.3 hours post fertilization, hpf) were exposed for 24 h to either constant control temperature alone (27°C), or to treatments consisting of thermal stress (19 peaks of sublethal 32°C), 24 h of constant temperature followed by UV radiation damage/repair assay [23], or both (first 24 h of heat peaks followed by UV radiation damage/repair assay). The UV assay consisted of 6 min of UVB followed by 15 min of UVA. Embryos were subsequently humanely sacrificed by snap-freezing at -80°C, followed by RNA extraction, cDNA synthesis, and Illumina sequencing [66]. Differential gene expression was performed with DESeq2 v1.28.1 [91] following read quality control using *fastp* v0.23.1 [92] and mapping against the zebrafish reference genome with the STAR v2.6.1 aligner [93]. To identify differentially expressed genes (DEGs), we compared various treatment comparisons to a control (n = 3 pools of 20 embryos *per* treatment). We abbreviated each comparison for clarity: "TS'' represents the comparison between the thermal stress treatment and the control, "UV" denotes the UVB/UVA treatment to the control, and "TS->UV" refers to the comparison of thermal stress followed by UVB/UVA damage/repair assay against UVB/UVA damage/repair assay without prior thermal stress. DEGs were determined to be significant when $p$-adj < 0.05. This paper, compared to Feugere and colleagues [12,66], outlines gene expression differences as a result of thermal stress and UV exposure, and specifically characterizes network statistics using experimental data.

The position of a node in a network is dependent on the overall structure of the network, since statistical properties of experimentally-determined nodes are computed relative to all other nodes in a network. In this study, two reference networks are used: **(i)** The first reference network, a complete PPI network for zebrafish, was a filtered and re-annotated STRING network obtained from Fernando and colleagues [94] which contained 14,677 genes (nodes) and 247,439 interactions (edges), with an edge evidence score of 0.9 out of 1, hereafter referred to as the zebrafish interactome. This PPI was lacking 1006 genes (nodes) from our experimentally expressed genes. Of these, 873 genes and their interactions (edges) were manually imported through STRING (https://string-db.org/). The missing genes were fitted into the network through matching the connecting genes found in the original network via.sif format. (ii) Since our experimental data was obtained from a transcriptomics experiment, only genes that are expressed in the developing embryo around 24 hpf would be likely to perform functional interactions with one another and may therefore constitute another appropriate reference network. The second reference network was therefore the final interactome, but filtered by all genes which were not expressed in any of our experiments with a read count of >10. This is here referred to as the expressed network. This network contained 10,953 nodes and 159,302 edges. For each reference network, any other single, unconnected nodes were removed from the analysis as their network properties would not be known.

Within any PPI network, nodes represent proteins, and edges connect interacting proteins. We previously demonstrated that the topology of nodes within a network can be characterized by decomposing a node's position into the three statistical parameters mentioned in the results section: Neighborhood connectivity (NC), Average Shortest Path Length (ASPL), and Betweenness Centrality (BC) assigns positioning of nodes within the network [34]. These three network statistical parameters were calculated for all nodes in the reference networks and all DEGs from TS, UV, and TS->UV overlaid over these, using the Network Analyzer function in Cytoscape (v.3.10.0, [34,95]). These three metrics were then used to bin each node into one of three categories using Discriminant Function Analysis (DFA). All genes in the network and DEGs were assigned to peripheral nodes (P), intermediate nodes (I) and hub nodes (H), by considering their individual ASPL, NC and BC values, respectively, as previously described in Wollenberg Valero's papers [34,38].

Not only do network node statistical parameters change based on the size and content of the reference network, but also the adjusted *p*-value of differential gene expression differs based on the number of comparisons performed (genes in a dataset). Therefore, for each comparison between an experimental condition and a reference network, the *p*-value for differential gene expression was adjusted based on the size of the reference network (either the interactome; or the expressed network), using the *fdr* function (RStudio v.3.4.0, [96]) to control for false positives.

Subsequently, DEGs in each of the experimental comparisons TS, UV, and TS->UV were overlaid over each of the two reference networks in Cytoscape. The following sets of comparisons were made: (i) between the reference interactome and the stress GO term genes; (ii) between the reference interactome and DEGs from the experimental data; (iii) the expressed network and the significant experimental DEGs. The data were not normally distributed (all comparisons *p*<0.001) according to Kolmogorov-Smirnov tests. Statistical tests were performed with SPSS v.27 [97].

To test hypothesis 1, the zebrafish interactome was compared to the stress GO term network, using non-parametric Kruskal-Wallis tests followed by pairwise Mann-Whitney Wilcoxon tests. Network statistics were generated, and networks plotted in Cytoscape. Due to the high number of genes per comparison, we calculated the partial ή2 for Cohen's h statistic effect size [98]. The effect size was small if it was<0.06; moderate if between 0.06 and 0.14; and large if ≥ 0.14.

To test hypothesis 2 whether the transcriptomic responses to each treatment comparison in developing zebrafish are constrained to specific parts of the PPI, Kruskal-Wallis tests followed by pairwise Mann-Whitney Wilcoxon tests between network statistical parameters of DEGs vs. the interactome or the expressed network were used and adjusted by Bonferroni correction. Partial ή2 for Cohen's h statistic effect size was calculated throughout, using Cohen's 1988 interpretation classification [99].

We then tested hypothesis 3, by selecting synergistic (combined treatment) and antagonistic DEGs (only present in both single stressor comparisons) and comparing their network statistical properties to the interactome and expressed network, respectively as described above.

To test hypothesis 4, the significant DEGs per treatment comparison (TS, UV, TS->UV) were assessed for biological pathways, phenotype and diseases using the ShinyGO (v.8.0) features '*GO biological processes*', '*ZFIN phenotype*' and '*ZFIN diseases*' respectively [100] and compared based on their location within the zebrafish interactome. The full listing of each associated phenotypic feature is listed in the supplementary data (S1 Dataset, Biological processes - Fig 7A; 6. Phenotype - Fig 7B; 7. Diseases - Fig 7C). DEG features shared by treatment comparisons were visualized using chord diagrams using *circlize* [101]. Patterns of association between node categories or treatment comparisons were identified using chord diagrams and ShinyGO data. Graphs were prepared in R (v.4.3.0) using *Eulerr* [102,103], *plotly* [104] for 3D scatter plots, *ggplot2* [105] for error plots, and *circlize* [101] for chord diagrams. *ZFIN* phenotype and disease features can give insights on the normal functioning of genes, despite that the data is obtained from cases where the gene is malfunctioning or mutated [106,107]. In order to integrate these ShinyGO features with experimentally-measured phenotypic outcomes in zebrafish embryos responding to the three treatment comparisons, data on sum of malformations and shortest embryo length were collected from 4 days post fertilisation (dpf) larvae [12]. Malformation scores were the sum of the presence or absence of three malformation types: bent spine, tail malformation, and pericardial edema. Shortest embryo length (n>29–30 *per* treatment) was the shortest distance from the head to the tail of the embryo measured with NIH ImageJ [108]. These phenotypic responses were, together with NC, ASPL and BC, standardized from -1 to 1 to be able to compare their relative changes across treatment comparisons.

## Supporting information

**S1 Fig. The position of synergistic and antagonistic DEGs within the zebrafish interactome.** (A) Distribution of interactome network nodes within network parameter space represented by NC, ASPL, and BC, with node categories shown in green (P nodes), blue (I nodes), and purple (H nodes). (A) Antagonistic genes and (C) Synergistic genes are shown in yellow, respectively. Inset pie charts represent node category distribution (large - interactome; small/top -

antagonistic, small/bottom - synergistic genes). (B,D) Error plos of ASPL, BC and NC for the interactome and synergistic and antagonistic gene sets' network dimensions. Pairwise Mann-Whitney U tests were not significant.
(EPS)

**S1 File. Supplementary tables (Tables A-K).**
(DOCX)

**S1 Dataset. containing raw data for tables and figures in manuscript.**
(XLSX)

## Acknowledgments

We acknowledge the Viper High Performance Computing facility of the University of Hull and its support team.

## Author contributions

**Conceptualization:** Katharina C Wollenberg Valero.

**Data curation:** Kaylee Beine.

**Formal analysis:** Kaylee Beine, Lauric Feugere, Alexander P Turner, Katharina C Wollenberg Valero.

**Funding acquisition:** Katharina C Wollenberg Valero.

**Investigation:** Kaylee Beine, Lauric Feugere, Katharina C Wollenberg Valero.

**Methodology:** Lauric Feugere, Katharina C Wollenberg Valero.

**Project administration:** Katharina C Wollenberg Valero.

**Software:** Alexander P Turner.

**Supervision:** Katharina C Wollenberg Valero.

**Visualization:** Kaylee Beine.

**Writing – original draft:** Kaylee Beine, Lauric Feugere, Alexander P Turner, Katharina C Wollenberg Valero.

**Writing – review & editing:** Kaylee Beine, Lauric Feugere, Alexander P Turner, Katharina C Wollenberg Valero.

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
