## [Decision Letter · Decision Letter 0]

PCOMPBIOL-D-24-01252

Network architecture of transcriptomic stress responses in zebrafish embryos

PLOS Computational Biology

Dear Dr. Wollenberg Valero,

Thank you for submitting your manuscript to PLOS Computational Biology. First of all, please excuse the exceptional long handling time for your manuscript which was due to a change in the section editor handling the manuscript after submission and an unusual difficulty to find a suitable academic editor and afterwards reviewers for this manuscript. To not further prolong the process, I have decided to make a decision based on a single review while we typically require three. I'm deeply sorry for the long time this required. After careful consideration, we feel that your manuscript has merit but does not fully meet PLOS Computational Biology's publication criteria as it currently stands. Therefore, we invite you to submit a revised version of the manuscript that addresses the points raised during the review process.

Please submit your revised manuscript within 60 days Apr 28 2025 11:59PM. If you will need more time than this to complete your revisions, please reply to this message or contact the journal office at ploscompbiol@plos.org. Please include the following items when submitting your revised manuscript:

We look forward to receiving your revised manuscript.

Kind regards,

Christoph Kaleta

Section Editor

PLOS Computational Biology

**Journal Requirements:**

At this stage, the following Authors/Authors require contributions: Kaylee Beine, Lauric Feugere, Alexander P Turner, and Katharina C Wollenberg Valero. Please ensure that the full contributions of each author are acknowledged in the "Add/Edit/Remove Authors" section of our submission form.

1) If the funders had no role in your study, please state: "The funders had no role in study design, data collection and analysis, decision to publish, or preparation of the manuscript."

**Reviewers' comments:**

Reviewer's Responses to Questions

Reviewer #1: Beine et al. examined the topology of stress response networks in zebrafish embryos. They utilized transcriptional data to find experimental evidence for network conservation in response to stressors. They proposed and tested four hypotheses through analyses such as network comparison and pathway enrichment. Overall, the study is compelling, and the findings are significant. However, the manuscript would benefit from clearer explanations of the results and a more concise, precise summary of the key findings in the abstract.

More detailed comments are as follows:

The full names of terms should be introduced before using their acronyms, such as ASPL, BC, NC and KW-H. An explanation of how each of these metrics contributes to node centrality should be included.

Figure 3, 4, 5 demonstrate that stressors employ distinct strategies to achieve centrality. For example, in Figure 3, TS had similar ASPL but a lower NC and higher BC. UV had a significantly lower ASPL and no significant difference in NC and BC. The authors should include an explanation or deeper exploration on these different network organization strategies.

The conclusion, "Combined stressors activate more central nodes while stressor-unique DEGs are more peripheral," is not clearly supported by the results. While the text mentions a slight increase in H nodes for combined stressors, no significant differences in ASPL, BC, or NC are reported. Statistical metrics quantifying these centrality changes should be included to substantiate the findings. For UV single stressors, the activation of more I nodes and fewer P nodes, along with lower ASPL and BC, is noted. However, the authors should clarify the criteria for defining these changes as “more peripheral” and ensure their significance is adequately supported.

The authors described the co-variation of DEG network parameters and phenotypic outcomes in lines 260–267. Adding a discussion on these results would provide clearer insights into how network topology/metrics influences phenotypic outcomes.

In Figures 1, 2-4, the axes for ASPL, NC, and BC should have consistent scales across to avoid confusion. For example, in the middle panel of Figure 1B, the difference in NC appears dramatic, but the statistical test indicates it is insignificant. The y-axis scale in the left and right panels also exaggerate the differences in ASPL and BC, making them appear more significant than they are. Also, there is an error in the text – “higher but not significantly different NC”, which is not consistent with the result in the figure.

In Figure 2B, the Venn diagram should represent the size of different DEG numbers proportionally to provide a more accurate visual representation of their relative sizes.

In figure 5A, overlay the yellow dots on top to better compare the distributions.

Figure 7 misses color legend for green, light green, dark green, and turquoise, making it difficult to identify the colors from their names accurately.

Avoid using informal language such as “usually,” “mostly,” and “some” when describing scientific results (e.g., lines 28–29, line 42).

Line 132, Protein-Protein Interaction (PPI) → PPI

Line 264�SEL is the sum of embryo length? “ASPL NC” —> “ASPL and NC”

**Have the authors made all data and (if applicable) computational code underlying the findings in their manuscript fully available?**

Reviewer #1: **No: **

PLOS authors have the option to publish the peer review history of their article (what does this mean? ). If published, this will include your full peer review and any attached files.

**Do you want your identity to be public for this peer review?** For information about this choice, including consent withdrawal, please see our Privacy Policy .

Reviewer #1: No

**Figure resubmission:**
---

## [Decision Letter · Decision Letter 1]

Dear Dr. Wollenberg Valero,

We are pleased to inform you that your manuscript 'Network architecture of transcriptomic stress responses in zebrafish embryos' has been provisionally accepted for publication in PLOS Computational Biology.

Best regards,

Christoph Kaleta

Section Editor

PLOS Computational Biology

Christoph Kaleta

Section Editor

PLOS Computational Biology

Reviewer's Responses to Questions

**Comments to the Authors:**

Reviewer #1: Overall, the revisions have strengthened the manuscript. The changes improve interpretability, readability, and the connection between network features and biological outcomes. I am satisfied with your responses and the revised version and support the manuscript’s progression toward publication.

**Have the authors made all data and (if applicable) computational code underlying the findings in their manuscript fully available?**

Reviewer #1: Yes

PLOS authors have the option to publish the peer review history of their article (what does this mean? ). If published, this will include your full peer review and any attached files.

**Do you want your identity to be public for this peer review?** For information about this choice, including consent withdrawal, please see our Privacy Policy .

Reviewer #1: No

---

## [Editor Report · Acceptance letter]

PCOMPBIOL-D-24-01252R1

Network architecture of transcriptomic stress responses in zebrafish embryos

Dear Dr Wollenberg Valero,

I am pleased to inform you that your manuscript has been formally accepted for publication in PLOS Computational Biology. Your manuscript is now with our production department and you will be notified of the publication date in due course.

With kind regards,

Lilla Horvath
